# Analytical Determination of Static Deflection Shape of an Asymmetric Extradosed Cable-Stayed Bridge Using Ritz Method

**DOI:** 10.3390/ma15124255

**Published:** 2022-06-15

**Authors:** Danfeng Lou, Yong Chen, Qian Feng, Jinbiao Cai

**Affiliations:** 1Faculty of Urban Construction, Zhejiang Shuren University, Hangzhou 310015, China; danfenglou@zjsru.edu.cn; 2Shanghai Research Institute of Building Sciences Co., Ltd., Shanghai 201108, China; zjuchenyong@163.com; 3Faculty of Civil Engineering and Architecture, Zhejiang University, Hangzhou 310058, China; fengqian@zju.edu.cn

**Keywords:** asymmetric extradosed cable-stayed bridge, mechanical behavior analysis, Ritz method, parametric study, composite structures

## Abstract

A practical method to analyze the mechanical behavior of the asymmetric extradosed cable-stayed (AECS) bridge is provided in this paper. The work includes the analysis of the equivalent membrane tension of the cables, the ratio of side-span cable force to middle-span cable force, and the deflection of the main girder subject to uniformly distributed load. The Ritz method is a simple and efficient way to solve composite structures, such as the AECS bridge, compared with the traditional force method, displacement method, or finite element method. The theoretical results obtained from the Ritz method are in good agreement with that from the finite element analysis, which shows the accuracy of this approach. Then, a parametric study of AECS bridges is carried out by using the proposed equations directly, instead of using the traditional finite element modeling process, which requires a lot of modeling work. As a result, reasonable values of very important parameters are suggested, which helps the readers reach a better understanding of the mechanical behavior of AECS bridges. More importantly, it helps the designers to enhance the efficiency in the stage of conceptual design.

## 1. Introduction

The extradosed cable-stayed (ECS) bridge is a relatively new composite structure consisting of stay cables, a tower, and a girder. The concept, first proposed by the French engineer Jacques Mathivat in 1988 [1,2] is based on the fact that the stay cables provide prestress for the girder and simultaneously share the vertical load. The concept was well received in Japan and more than 30 ECS bridges were constructed there [3,4,5,6,7]. In the meantime, this type of structure was widely used. More and more ECS bridges were constructed in Korea [8], India [9], Hungary [10], Thailand [11]), Slovenia [12], America [13], etc., due to the consideration of aesthetic configurations with lower towers and the reduction of cost. On this basis, Meng et al. [14] presented a new type of combined cable-stayed bridge called the extradosed and intradosed cable-stayed bridge with continuous cables, which optimized the structure performance of ECS bridges.

The tower of the ECS bridges is relatively low and the stiffness of the girder is large, compared with the normal cable-stayed bridges. As a result, the structural behavior of the ECS bridges is partly like that of cable-stayed and partly like girder bridges [15]. Liu et al. [16,17] had defined some parameters based on the influence of load effect on cables. It helps to explore the boundaries of this form of bridge, which can comprehensively reflect the mechanical properties of ECS bridges. Chen [18,19] presented the concept of cable/beam live-load ratio to identify the mechanical behavior of this bridge. However, for the hybrid structure, the guidance of structural design to reflect the internal action of the elements is lacking, due to the variety of components and high degree of static indeterminacy. Yi et al. [20] derived the girder deflection formula of the single span beam-arch bridge based on the force method. Furthermore, Chen et al. [21,22] presented the formula of the load shared ratio between different components of the three-span continuous beam-arch bridge and suspension bridge based on the force method. Both of them provided a reliable approach to analyze the structure behavior of the hybrid system. However, the process for solving the force method was indeed complex because of the vast number of equations. Lonetti et al. [23] proposed an optimization design method for hybrid cable-stayed bridge based on a two-step algorithm. The finite element analysis was then provided by Dou et al. [24] and Yi et al. [25] for introducing the design procedures and parametric studies of the ECS bridges. Moreover, the dynamic property of cable-stayed bridges was also studied by researchers. Kim et al. [26] identified the modal damping ratios from the operational monitoring data to assess the vibrational serviceability performance of a parallel cable-stayed bridge. Xu [27] investigated the seismic performance of a cable-stayed bridge with passive energy dissipation devices. 

According to previous studies, most of the existing ECS bridges are designed and analyzed by using traditional techniques. Empirical or semi-empirical iterative methods are utilized by designers. However, some research efforts are carried out to propose available procedures to achieve the interactions between different components and simplify the conceptual design procedure. In particular, the author [28] applied the variational principle to derive the girder deformation equations of the symmetrical ECS bridge. Then Su et al. [29] set forth the method to analyze the girder bending moment, which provides a new way to study the mechanical performance of this type of bridge. Currently, more and more asymmetric extradosed cable-stayed (AECS) bridges have being constructed. Due to the different performance of structural behaviors caused by the geometric asymmetry, further investigation is required in order to achieve a better understanding and designing of the structure. Accordingly, this paper aims to analyze the mechanical behavior of AECS bridges based on the variational principle using the Ritz method, which is the first time this method has been applied to studying the structural performance of AECS bridgse. Practical equations of the equivalent membrane tension of the cables, the ratio of side-span cable force to middle-span cable force (hereafter referred to as cable force ratio), and girder deflection of the AECS bridge are developed by using the Ritz method. It is a simple way to solve the variational problem, compared with the traditional force method, the displacement method, and the finite element analysis. The results were compared with the finite element analysis with good agreement, which shows the accuracy of the application of the Ritz method to the analysis of the AECS bridge. A parametric study is then presented to analyze the influence of different components on the structure behavior, based on which the reasonable ranges of the parameters in the conceptual design stage are also provided. The analyses show that the Ritz method, as an approximate solution based on the variational principle, is an effective way to simulate the structural behavior of the AECS bridge. It also helps the designers make quick decisions during the conceptual design stage.

## 2. Methodology

The Ritz method is a direct method to solve the variational problem. The variational problem focuses on the extreme value of the functional. Generally speaking, a functional is a function whose argument is a class of functions. The calculus of variations has found wide application in mathematical physics. Shi et al. [30] used energy-variation principles to analyze the static characteristics of a multi-rib T-beam under the impacts of shear lag effect. Zhang [31] studied the rail continuous bending of ballastless track in the high-seed railway based on the energy variation principle. Hence, by applying the variational principle to the analysis of AECS bridges creatively, several theoretical formulas could be obtained with simple steps. According to the Ritz method, the determining of the function that makes the variational functional extreme is based on the representation of the unknown function in the functional by a suitable series with constants or functions, which can be obtained by a minimizing process. It is known that of all the kinematically admissible displacement fields the one that makes the potential energy of the structure minimum is the actual displacement field. Thus, in accordance with the principle of minimum potential energy, the deflection function of the girder must make the potential energy of the girder minimum.

When using the Ritz method, first of all, the girder deflection function is assumed as:$$y(x)=\sum_{i=1}^{n}c_{i}\varphi_{i}(x)$$
where the coefficients $c_{i}$ are to be determined and the sequence of functions $\varphi_{i}(x)$, referred to as the coordinate functions, is a preassigned complete sequence of functions that satisfy the boundary conditions.

Then, the potential energy of the system can be expressed as the functional of the assumed girder deflection function.
$$U=U[y(x)]=U[\sum_{i=1}^{n}c_{i}\varphi_{i}(x)]$$

The values of $c_{i}$ are determined through the minimizing conditions.
$$\frac{\partial U}{\partial c_{i}}=0\;i=1,2\cdots n$$

The problem is solved by substituting the values of $c_{i}$ into the assumed deflection function.

### 2.1. Engineering Background

The Lingjiang Bridge is used as a basic model here. The main bridge is a single pylon AESC bridge with a span of (76 m + 91.2 m). Eleven parallel stay cables are arranged on a single plane, with cross-sectional area $A_{s}=1.036\times 10^{-2}\;m^{2}$ and elastic modulus $E_{s}=1.95\times 10^{5}$ MPa. The angle of the stays is $\theta =14^{\circ }$ and the interval between stay cables on the main girder is $d_{1}=4\;m$, while that on the tower is $d_{2}=1\;m$. The main girder is made of concrete, with the cross-sectional area $A_{b}=20.8\;m^{2}$, the elasticity modulus $E_{b}=3.45\times 10^{4}\;\mathrm{MPa}$, and the moment of inertia $I_{b}=23.96\;m^{4}$. The tower height above the bridge deck is 16 m and pier height below the bridge deck is 15 m. The tower and the pier are consolidated together while the girder is supported by the pier. The length of the free cable area on the tower is 6 m, and the length of the cable area is 10 m. The elastic modulus of the tower is $E_{b}=3.45\times 10^{4}\;\mathrm{MPa}$ and the moment of inertia is $I_{t}=10.67\;m^{4}$. Figure 1 shows the configuration of the basic model. The numerical analyses were performed under the uncracked concrete condition.

### 2.2. Assumptions

The ECS bridge is mainly affected by dead load and vehicle live load. The uniformly distributed load is a type of classical basic load which has been considered here to deduce the practical calculation formulas of the system. The computing procedures are conducted based on the following assumptions:

(1) As the stiffness of the girder is large and the deformation of the girder is slight, it is supposed that the material is considered linear elastic.

(2) The cable forces on each side of the tower are assumed as equal under the uniform load, and the vertical component of the cable forces could be treated as an equivalent membrane tension.

According to the FEM results in the Figure 2, it is found that the cable forces of per cables are almost around the average value. Moreover, Konstantakopoulos et al. [32] and Michaltsos et al. [33] proved that for a very dense distribution of the stay cables, the vertical component of the cable forces can be replaced with the distributed load. Chen [18] also analyzed the cable-beam live load ratio of the ECS bridge by assuming the vertical component of the cable forces as a uniform load. Yi et al. [20] treated the suspender forces of the beam–arch hybrid bridge as an equivalent membrane tension. Therefore, the membrane analogy is adopted here in the analysis of cable forces. 

(3) As the height of the pylon is low and the stiffness of the deck is large, the influence of the axial forces either of the tower or of the girder could be neglected [33,34]

(4) For the tower-pier consolidation support system, the girder can be regarded as a double span continuous beam. The simplified mechanic model is shown in Figure 3.

Based on the assumptions, when the structure is subjected to uniformly distributed load q, it can be assumed that the vertical component of cable force on the long span is t, and that on the short span is kt, where the coefficient k is called the ‘cable force ratio’ (which is here caused by the asymmetry of the spans). The length of the long span is defined as l, the length of the short span is ηl, where the coefficient η is the span ratio. The unsupported length near the tower is $l_{1}=al$, the supported length on the girder is $l_{2}=bl$, the unsupported length near the support on the long span is $l_{3}=cl$, and that on the short span is $l_{3}^{\prime }=c^{\prime }l$.

### 2.3. Formula Derivation

First of all, the finite element model (Figure 4) was obtained by Midas software. The girder of the model is composed of 84 beam elements, each element is two meters long. The cables are simulated by the cable elements, which can only provide tension forces. The pier and tower are also used beam elements. The geometric characteristics of the basic model in Figure 1 is employed in the finite element model. The external force is 1 kN/m. Hence, the deflection of the girder can be calculated by the finite element analysis, as shown in Figure 5. The deflection curves are fitted with the quartic polynomial perfectly, as shown in Figure 6.

According to the FEM fitting results, it can be assumed that the deflection expressions of the short and long span are:(1)$$y_{1}=a_{1}x_{1}+a_{2}x_{1}^{2}+a_{3}x_{1}^{3}+a_{4}x_{1}^{4}+a_{0}$$
(2)$$y_{2}=b_{1}x_{2}+b_{2}x_{2}^{2}+b_{3}x_{2}^{3}+b_{4}x_{2}^{4}+b_{0}$$
where $a_{0}\sim a_{4}$ and $b_{0}\sim b_{4}$ are undetermined coefficients.

(1) The potential energy of the system

The bending strain energy of the girder can be obtained based on the assumed deflection expressions:(3)$$U_{b}=\int_{0}^{\eta l}\frac{E_{b}I_{b}}{2}{\left({y_{1}}^{\prime \prime }\right)}^{2}dx_{1}+\int_{0}^{l}\frac{E_{b}I_{b}}{2}{\left({y_{2}}^{\prime \prime }\right)}^{2}dx_{2}$$

The bending potential energy caused by uniform load q and equivalent membrane tension of the stay cables are:(4)$$U_{q}=\int_{0}^{\eta l}qy_{1}dx_{1}+\int_{0}^{l}qy_{2}dx_{2}$$
(5)$$U_{t}=-\int_{(\eta -a-b)l}^{(\eta -a)l}kty_{1}dx_{1}-\int_{al}^{(a+b)l}ty_{2}dx_{2}$$

Then the total potential energy of the system is:(6)$$U=U_{b}-U_{q}-U_{t}$$

(2) Boundary conditions

For the short span, when $x_{1}=0$, $y_{1}=0$; $x_{1}=\eta l$, $y_{1}=0$; for the long span, when $x_{2}=0$, $y_{2}=0$; $x_{2}=l$, $y_{2}=0$, so that $a_{0}=b_{0}=0$, and
(7)$$a_{1}\eta l+a_{2}\eta^{2}l^{2}+a_{3}\eta^{3}l^{3}+a_{4}\eta^{4}l^{4}=0$$
(8)$$b_{1}l+b_{2}l^{2}+b_{3}l^{3}+b_{4}l^{4}=0$$

The rotation angle of the section and bending moment of short span and long span at the mid-span support are equal to $y_{1}{}^{\prime }{|}_{x_{1}=\eta l}=y_{2}{}^{\prime }{|}_{x_{2}=0}$, $y_{1}{}^{\prime \prime }{|}_{x_{1}=\eta l}=y_{2}{}^{\prime \prime }{|}_{x_{2}=0}$, and the results are as follows:(9)$$a_{1}+2a_{2}\eta l+3a_{3}\eta^{2}l^{2}+4a_{4}\eta^{3}l^{3}=b_{1}$$
(10)$$a_{2}+3a_{3}\eta l+6a_{4}\eta^{2}l^{2}=b_{2}$$

From Equations (7)–(10), the coefficients $a_{1},a_{2},a_{3},b_{3}$ can be expressed as follows:(11)$$a_{1}=-2b_{1}+\eta lb_{2}-\eta^{3}l^{3}a_{4}$$
(12)$$a_{2}=\frac{3}{\eta l}b_{1}-2b_{2}+3\eta^{2}l^{2}a_{4}$$
(13)$$a_{3}=-\frac{1}{\eta^{2}l^{2}}b_{1}+\frac{1}{\eta l}b_{2}-3\eta la_{4}$$
(14)$$b_{3}=-\frac{b_{1}}{l^{2}}-\frac{b_{2}}{l}-lb_{4}$$

(3) The condition for the minimization of the total potential energy functional of the system is:(15)$$\frac{\partial U}{\partial b_{1}}=0$$
(16)$$\frac{\partial U}{\partial b_{2}}=0$$
(17)$$\frac{\partial U}{\partial b_{4}}=0$$
(18)$$\frac{\partial U}{\partial a_{4}}=0$$

The expressions of the undetermined coefficients $a_{i}$ and $b_{i}$ of the two deflection functions $y_{1}(x)$ and $y_{2}(x)$ can be obtained by solving Equations (11)–(18):(19)$$a_{i}=\delta_{i1}+\delta_{i2}t+\delta_{i3}kt$$
(20)$$b_{i}=\xi_{i1}+\xi_{i2}t+\xi_{i3}kt$$
where $\delta_{ij}$ and $\xi_{ij}$(i=1,2⋯4, j=1,2,3) are constants determined by the values of a, b, η and the bending stiffness $E_{b}I_{b}$ of the main girder. The specific equations for $\delta_{ij}$ and $\xi_{ij}$ are presented in Appendix A.

(4) Deformation compatibility

The equation for deformation compatibility is based on the vertical displacement of girder at Point A and B, which are the locations of the two outermost stay cables, as shown in Figure 7.

By substituting $x_{1}=(\eta -a-b)l$ into $y_{1}(x)$ and substituting $x_{2}=(a+b)l$ into $y_{2}(x)$, the deflections of Point A and Point B are expressed as:(21)$$f_{1}=F_{11}+F_{12}t+F_{13}kt$$
(22)$$f_{2}=F_{21}+F_{22}t+F_{23}kt$$
where $F_{1j}$, $F_{2}{}_{j}$ are constants expressed as:(23)$$F_{1j}=\sum_{i=1}^{4}{\left(\eta -a-b\right)}^{i}l^{i}\delta_{ij}(j=1,2,3)$$
(24)$$F_{2j}=\sum_{i=1}^{4}{\left(a+b\right)}^{i}l^{i}\xi_{ij}(j=1,2,3)$$

Cable forces on the different sides of the tower are not equal due to the asymmetry of the span length. The tower will produce a certain longitudinal horizontal displacement to the long span under the effect of unbalanced cable forces. By assuming that the cable forces are the equivalent membrane tension mentioned before, the horizontal component of cable forces acting on the tower can be treated as a uniform load as well. The tower can be simplified as a cantilever beam. According to the physical parameters defined before, the interval of the cables on the girder is $d_{1}$, while that on the tower is $d_{2}$, the angle of the cable is θ, the total height of the tower and pier is h, the height supported by the cables is $e_{t}h$, so that the equivalent force diagram of the tower is shown in Figure 8.

Due to the mechanical deformation law, the horizontal displacement at the top of the tower is expressed as follows:(25)$$f_{t}=\frac{(1-k)t\cot^{2}\theta h^{4}}{24E_{t}I_{t}}(8e_{t}-6e_{t}^{2}+e_{t}^{4})$$

Obviously, the vertical deflection at the anchorage point of the outermost stay cable is caused by the extension of the cable and the longitudinal horizontal displacement of the tower. Hence, the vertical deflections of Point A and B can be also expressed as shown below due to their geometric relationship:(26)$$f_{1}=\frac{kt(a+b)l}{ea_{s}\sin^{2}\theta \cos\theta }-f_{t}\cot\theta$$
(27)$$f_{2}=\frac{t(a+b)l}{ea_{s}\sin^{2}\theta \cos\theta }+f_{t}\cot\theta$$
where $ea_{s}=\frac{E_{s}A_{s}}{d_{1}}$.

According to Equations (23) and (24) and Equations (26) and (27), the deformation compatibility equations can be obtained:(28)$$\left\{ \begin{array}{l} \frac{kt(a+b)l}{ea_{s}\sin^{2}\theta \cos\theta }-f_{t}\cot\theta =F_{11}+F_{12}t+F_{13}kt\\ \frac{t(a+b)l}{ea_{s}\sin^{2}\theta \cos\theta }+f_{t}\cot\theta =F_{21}+F_{22}t+F_{23}kt \end{array}\right.$$

The formulas of the equivalent membrane tension of cables t and cable force ratio k can be set by solving Equation (28):

(29)$$t=\frac{F_{21}(M_{1}+M_{2}-F_{13})+F_{11}(M_{2}+F_{23})}{\left(M_{1}+M_{2}-F_{13})(M_{1}+M_{2}-F_{22})-(M_{2}+F_{23})(F_{12}+M_{2}\right)}$$(30)$$k=\frac{F_{21}(M_{2}+F_{12})+F_{11}(M_{1}+M_{2}-F_{22})}{F_{21}(M_{1}+M_{2}-F_{13})+F_{11}(M_{2}+F_{23})}$$
where $M_{1}$, $M_{2}$ are constants: $M_{1}=\frac{(a+b)l}{ea_{s}\sin^{2}\theta \cos\theta }$, $M_{2}=\frac{\cot^{3}\theta h^{4}}{24E_{t}I_{t}}(8e_{t}-6e_{t}^{2}+e_{t}^{4})$.

By substituting Equations (29) and (30) into Equations (19) and (20), the values of coefficients $a_{1}\sim a_{4}$ and $b_{1}\sim b_{4}$ are determined. In other words, the expressions of the girder deflection of $y_{1}(x)$ and $y_{2}(x)$ are developed.

## 3. Formula Verification and Error Analysis 

The values of the equivalent membrane tension of stay cables t, the cable force ratio k, and the deflection of two spans can be calculated by substituting the geometric data of the basic model in Figure 1 into the formulas derived before (hereinafter referred to as Ritz method formulas). Meanwhile the finite element model was also calculated with the purpose of verifying the results of the Ritz method formulas. A comparison of the results of the two methods is shown in Table 1 and Figure 9. 

From the comparison results, it can be seen that the deflection of the girder obtained using the Ritz method formula is very close to that calculated by the finite element method, which shows that the formula derived in this paper is completely reasonable for solving the deflection of the girder with quite high accuracy. Furthermore, the error of cable force ratio k is 2.66%, which also meets the accuracy requirements. However, the calculation error of stay cables is slightly too large. In order to improve the accuracy of the analogy of the equivalent membrane tension, the reasons for the error will be analyzed, and the correction of the formula carried out. Firstly, this part of the error could be caused by the assumption that the cable forces on each side of the tower are the same and their vertical components are equivalent to a uniform load. However, the actual force of each cable is not equal, and it is a group of concentrated loads with the same interval. Moreover, as an approximate solution, the axial deformations of the girder and the tower, as well as the flexural strain energy in the tower are not considered here, which also cause the error. Moreover, human errors could exist within the finite element analysis. As the axial and flexural deformations of the tower are neglected, it implies that the stiffness of the tower is larger than its actual stiffness. It means the stiffness of the cables is smaller than the reality in turn. Thus, when we perform error correction, the stiffness of the cables needs to be modified in order to make the results more practical. By investigating all the coefficients in the formulas, we find that M_1_ mainly relates to the stiffness of the cables and has a great effect on the accuracy of the equivalent membrane tension. As a result, $M_{1}$ is modified by $M_{1}^{\prime }=\frac{M_{1}}{1.5}$ and the error of equivalent membrane tension is reduced to −5.1%, as shown in the first row of Table 2, which helps to meet the accuracy requirements.

The structural parameters, such as cross-sectional area of the cables, the angle of the cables, and the bending stiffness, are changed, and the remaining eight models are obtained in order to achieve further verification of the correctness and generality of the Ritz method formulas. The calculation results and errors compared to the FEM of each model are shown in Table 2, where $f_{a}$ is the midspan displacement of the short span, while $f_{b}$ is that of the long span.

According to the results, the errors of the cable force ratio k are all within 8%, and the absolute values of equivalent membrane tension error are all within 9%. The average error of $f_{a}$ is −6.61%, while that of $f_{b}$ is −3.82%. It is concluded that the accuracy of the energy method formulas meets the design requirements, especially in the conceptual design stage.

## 4. Parametric Studies

Instead of iterative finite element analysis, several important design parameters are studied by using the Ritz method formulas proposed previously. The investigation includes the unsupported length on the girder, span ratio, and angle of the stay cables, which all have a great effect on the mechanical behaviors of AECS bridges. The investigation in this part is based on the basic model in Figure 1, while the initial span layout is changed to (72 m + 90 m) in order to facilitate the analysis.

### 4.1. Length of the Unsupported Girder

When analyzing the influence of the length of the unsupported girder near the tower $l_{1}$, the span length l, the span ratio η, and the inclination angle of the cables stay the same, while $l_{1}$ varies from 8m to 48m by reducing the inner cables. Meanwhile, the cross-sectional area of per cable is increased to keep the total stiffness of the cable’s constant. A series value of the equivalent membrane tension of the cables t, the cable force ratio k, and the deflections of the mid-span of the two spans $f_{a}$ and $f_{b}$ are calculated when $l_{1}$ varies. Figure 10 shows the relationships between them, determined using the Ritz method formulas.

Figure 10a shows that with the increase of $l_{1}$, the value of t increases, while the value of k increases at first and then decreases. Figure 10b shows that both mid-span deflections $f_{a}$ and $f_{b}$ first decrease and then increase. An increase in the unsupported length near the tower implies a decrease in the relative stiffness of the girder. Consequently, the load shared by the girder is decreased and the equivalent membrane tension of the stay cables increases correspondingly. In turn, this leads to a reduction of the girder deflections. However, when the unsupported length exceeds a certain limit, the deflections of the mid-span will increase again because of the reduction of the cable supports. Thus, the unsupported length near the tower should have an optimum value. According to the results shown in Figure 10, it can be concluded that the optimum range of the unsupported length near the tower is $l_{1}$ = (0.35~0.40)l, both with small deflections and relatively balanced cable forces.

Similarly, while the span length l, the span ratio η, and the inclination angle of the cables stay the same, $l_{3}$ varies from 18 m to 58 m by reducing the outmost cables. To maintain the total stiffness of the cables, the cross-sectional area of per cable is also increased. The change of t, k, $f_{a}$, and $f_{b}$ relating to $l_{3}$, is shown in Figure 11.

Comparing Figure 11 with Figure 10, it can be seen that the variation trends of t, k, $f_{a}$, $f_{b}$ are the same whether $l_{1}$ or $l_{3}$ is increased. However, the variation range is broader while the unsupported length near the support varies. This indicates that the unsupported length near the support has a greater influence on the behavior of AECS bridges than that near the tower. The change of the unsupported length mainly transforms the relative stiffness between the girder and cables. In consideration of the girder deflections as well as the cable force ratio, the optimum range of the unsupported length near the support can be determined as $l_{3}$ = (0.40~0.45)l.

### 4.2. Span Ratio

The span ratio is a significant design parameter in the case of an asymmetric bridge. If the value of the long span reaches a certain amount, a negative support reaction will appear on the side of the short span. Hence some extra counterweight is needed to balance the force. Meanwhile, there will be an excessive bending moment on the long span, which has a negative effect on the structure. Therefore, it is necessary to determine a reasonable span ratio in the conceptual design stage. When analyzing the span ratio, the length of the short span remains 72 m and the length of the long span varies from 136 m to 72 m. For different span ratios, the values of the equivalent membrane tension t, the cable force ratio k, and the mid-span deflections of the two spans $f_{a}$ and $f_{b}$ can be calculated using the Ritz method formulas. The trends in variation of each factor are shown in Figure 12.

It can be seen from Figure 12a that with the increase of η, the value of t decreases, which means the load carried by the cables decreases. The explanation is that when the length of the long span decreases, the stiffness of the long span increases, so that the load shared by the girder grows, while the load distributed by the stay cables decreases. On the other hand, the decrease of the long span length leads to the reduction of deflection in the middle of the long span (Figure 12b), and the cable forces trend to be equivalent (Figure 12a). When the span ratio is 1, the structure is symmetrical, so that the cable force ratio k=1 and the mid-span deflections of two spans are the same. There is an important phenomenon shown in Figure 12b: when the span ratio is smaller than 0.7, the deflection of the short span is negative. This means that the deflection is upward. This can cause a negative reaction force on the short span support and supplementary measures, such as setting a counterweight or auxiliary pier, should be taken. In that case, the optimization of the mechanic behavior of the bridge should be carried out, which means the cost will increase. Thus, it is suggested that for the ASCE bridges with similar material properties and geometric layout mentioned in this paper, the span ratio should be greater than 0.7. According to Chen et al. (2005), the average span ratio of existing two-span ASCE bridges is 0.83, which matches our results. In fact, the span ratio is affected by lots of factors. Hence the value ‘0.7’ provides a suggestion for designers in the conceptual design stage. 

### 4.3. Inclination Angle of Cables

Another vital parameter for this type of bridge is the height of the low tower. This is because it implies a relatively small cable inclination angle, compared with the ordinary cable-stayed bridge. According to the Ritz method formulas, the influence of cable angle θ on the equivalent membrane tension t, the cable force ratio k, mid-span deflections of two spans $f_{a}$, $f_{b}$ can be obtained, and the results are shown in Figure 13.

It can be seen from Figure 13a,b that with the increase of the cable angle, the value of t increases linearly, which means the vertical load shared by the stay cables increases. Accordingly, the vertical load distributed to the main girder decreases, the deflection of the girder decreases, and the deflection gap between two spans is closed. Therefore, the cable forces on the two sides of the tower tend to be balanced, the cable force ratio approaches 1. According to Kasuga’s data [5], the vertical load distribution ratio of the stay cables is less than 30%. Hence, in accordance with Figure 13a, the reasonable angle of the cables is less than 25°. To sum up, the range of the cable angel suggested in this paper is generally between 15° and 25°, which fits well with that has been suggested by Chen et al. [18]. 

## 5. Conclusions and Outlook

A practical methodology, the Ritz method, is presented here to study the mechanical behavior of the AECS bridges. This is the first time this method has been applied to the analysis of the structural behavior of AECS bridges. Compared with the traditional force method, displacement method, or finite element method, the present method is much more efficient for solving the equations with relatively simple steps. Furthermore, the results of this paper help designers to avoid the tedious finite element modeling process in the conceptual design stage and determine the structural parameters more reasonably and efficiently. From the results, some vital conclusions could be drawn as follows:

(1) Theoretical equations of the equivalent membrane tension of the cables, the cable force ratio and the deflection of the girder under uniform load are established by using Ritz method. It was proven that the accuracy of the derived formulas satisfies the requirement of conceptual design. 

(2) The analysis of the unsupported length of the girder shows the length of the unsupported girder has a direct effect on the stiffness of girder, which affects the deflection of the girder and the load percentage carried by the cables. Moreover, the unsupported length near the support has a greater influence on the structure than that near the tower. For the AECS bridges with similar material properties in this paper, the reasonable range of the unsupported length near the tower would be around $l_{1}$ = (0.35~0.40)l, while that near the support is $l_{3}$ = (0.40~0.45)l.

(3) The span ratio of AECS bridges mainly influence the balance of the cable force on the different sides of the tower, which in turn has an effect on the deflection gap of the two spans. Based on our investigation, for AECS bridges with similar material properties and geometric layout to that described in this paper, the span ratio should be greater than 0.7 to avoid reserving the deflection of the short span. 

(4) The angle of the stay cables affects the percentage of the load carried by the stay cables significantly. With the increase of the angle, the vertical load distribution ratio of the cables increases almost linearly, which in turn leads to the decrease of the deflection of the girder. The suggested range of the cable angle is between 15° and 25° based on the results in this paper, combined with the consideration of the current construction data.

It can be expected that the Ritz method from the variational principle is also suitable to calculate other composite structures, such as suspension bridges and hybrid cable-stayed suspension bridges. Moreover, in addition to solving the deflection shape under uniform load, the Ritz method could also be applied to calculating other load conditions by changing the assumption of the deflection fiction. To sum up, the Ritz method has the potential for wide applications in the field of composite structure analysis.

## Figures and Tables

**Figure 1 materials-15-04255-f001:**
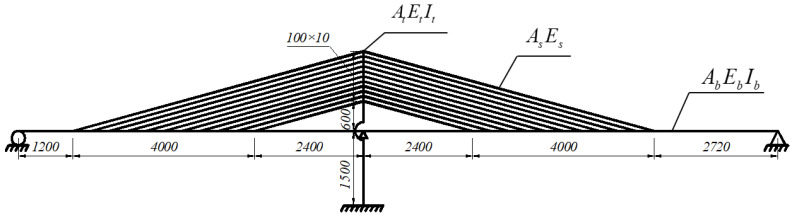
Configuration of the basic model (unit: cm).

**Figure 2 materials-15-04255-f002:**
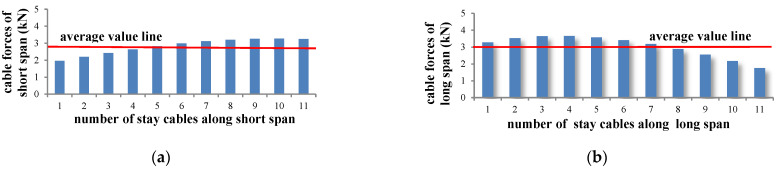
Cable forces obtained by FEM. (**a**) Cable forces of short span; (**b**) Cable forces of long span.

**Figure 3 materials-15-04255-f003:**
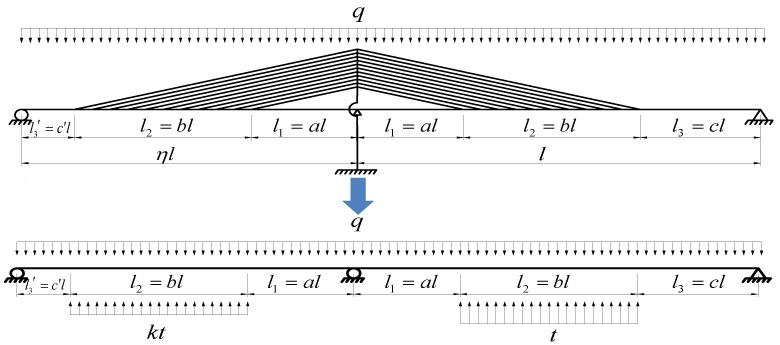
Simplified mechanical model.

**Figure 4 materials-15-04255-f004:**
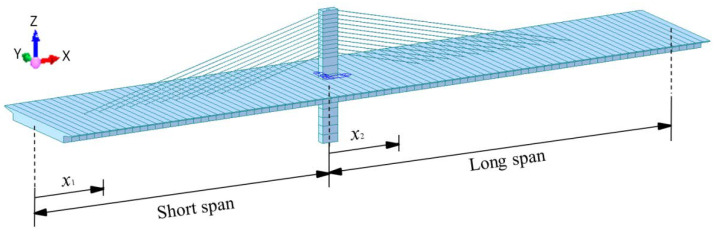
Finite element model of the structure.

**Figure 5 materials-15-04255-f005:**
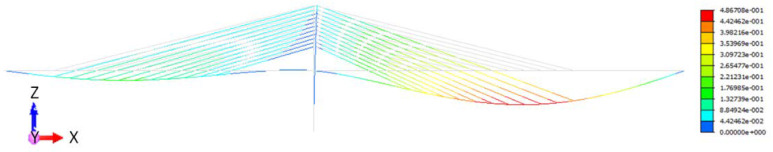
Deflection curves of the finite element analysis.

**Figure 6 materials-15-04255-f006:**
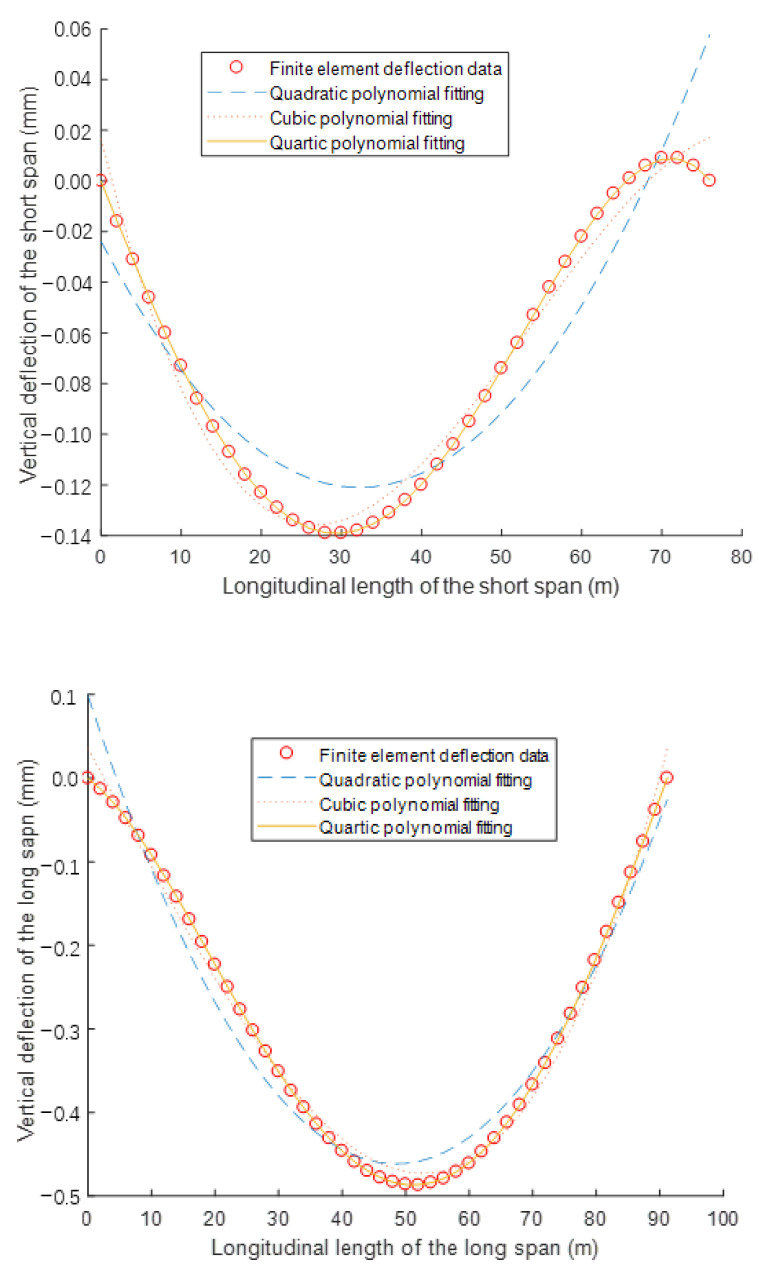
Fitting results of deflection curves.

**Figure 7 materials-15-04255-f007:**
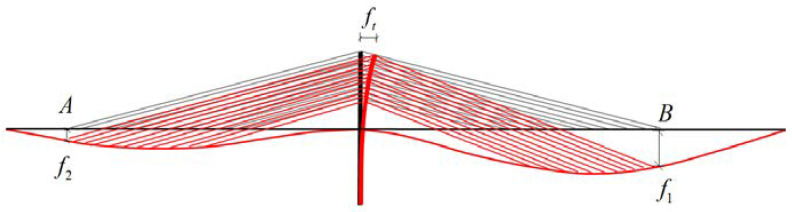
Deformation of the structure.

**Figure 8 materials-15-04255-f008:**
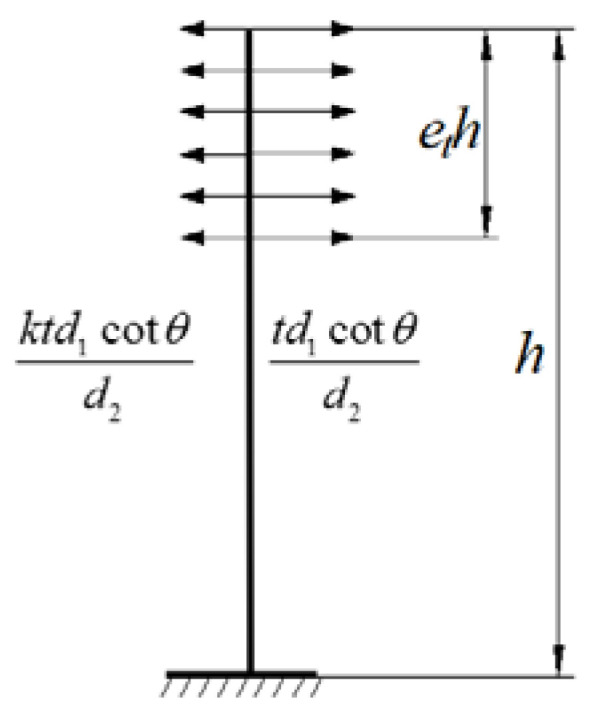
Equivalent force diagram of the tower.

**Figure 9 materials-15-04255-f009:**
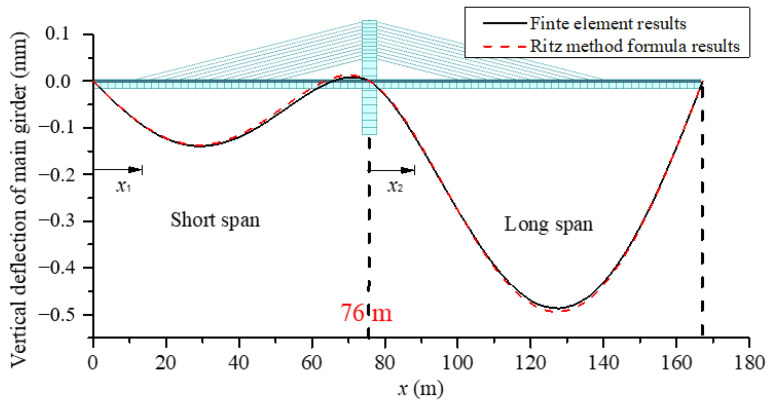
Comparison of girder deflections.

**Figure 10 materials-15-04255-f010:**
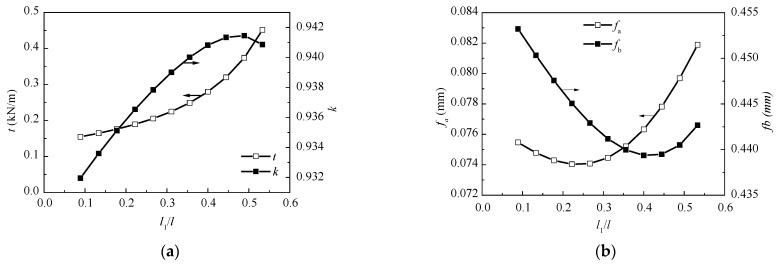
(**a**) Relationships between $l_{1}/l$ and t, k; and (**b**) the relationships between $l_{1}/l$ and $f_{a}$, $f_{b}$. The two curves correspond to different y coordinates, the arrow shows the y coordinate.

**Figure 11 materials-15-04255-f011:**
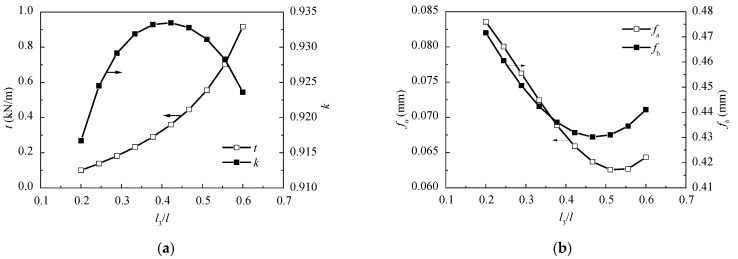
(**a**) Relationships between $l_{3}/l$ and t, k; and (**b**) the relationships between $l_{3}/l$ and $f_{a}$, $f_{b}$. The two curves correspond to different y coordinates, the arrow shows the y coordinate.

**Figure 12 materials-15-04255-f012:**
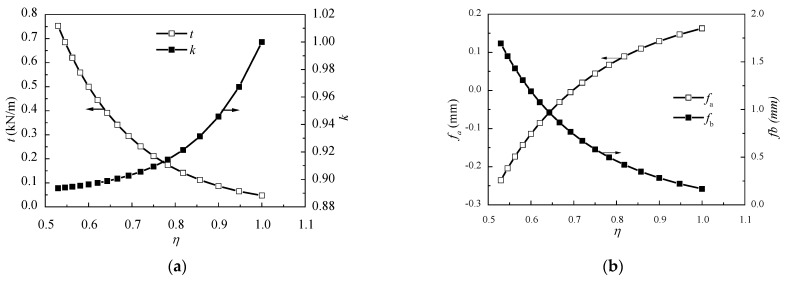
(**a**) Relationships between η and t, k; and (**b**) the relationships between η and $f_{a}$, $f_{b}$. The two curves correspond to different y coordinates, the arrow shows the y coordinate.

**Figure 13 materials-15-04255-f013:**
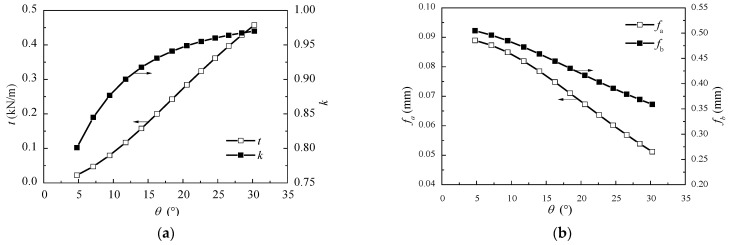
(**a**) Relationships between θ and t, k; and (**b**) the relationships between θ and $f_{a}$, $f_{b}$. The two curves correspond to different y coordinates, the arrow shows the y coordinate.

**Table 1 materials-15-04255-t001:** Comparison of the finite element method (FEM) and Ritz method formulas (RMF).

Basic Model	FEM	RMF	Error
k	0.925	0.900	−2.66%
t (kN/m)	0.186	0.124	−33.42%

**Table 2 materials-15-04255-t002:** Comparison of the finite element method (FEM) and Ritz method formulas (RMF).

Model Number	Parameters	k			t (kN/m)			$f_{a}(\mathrm{mm})$			$f_{b}(\mathrm{mm})$		
		FEM	RMF	Error (%)	FEM	RMF	Error (%)	FEM	RMF	Error (%)	FEM	RMF	Error (%)
1	Basic model	0.925	0.930	0.56	0.185	0.176	−5.10	0.086	0.080	−6.56	0.431	0.421	−2.28
2	$A_{s}=1/9A_{s0}$	0.650	0.632	−2.64	0.028	0.026	−8.43	0.100	0.092	−7.91	0.483	0.466	−3.51
3	$A_{s}=1/4A_{s0}$	0.784	0.781	−0.48	0.057	0.052	−8.47	0.097	0.090	−6.97	0.473	0.458	−3.16
4	$A_{s}=4A_{s0}$	0.973	0.980	0.71	0.490	0.511	4.26	0.057	0.053	−7.23	0.334	0.322	−3.72
5	$A_{s}=9A_{s0}$	0.984	0.990	0.67	0.748	0.804	7.49	0.032	0.029	−10.01	0.261	0.235	−10.15
6	tanθ=1/5	0.987	0.912	−7.63	0.124	0.121	−2.24	0.089	0.085	−5.04	0.453	0.437	−3.43
7	tanθ=1/2	0.989	0.969	−2.02	0.428	0.433	1.38	0.062	0.060	−3.82	0.355	0.344	−3.00
8	$I_{t}=8I_{t0}$	0.692	0.664	−4.00	0.210	0.201	−4.38	0.092	0.086	−6.26	0.419	0.408	−2.56
9	$I_{t}=1/8I_{t0}$	0.179	0.171	−4.75	0.987	0.990	0.31	0.084	0.079	−5.69	0.435	0.424	−2.60

## Data Availability

All data and models that support the findings of this study are available from the corresponding author upon reasonable request.

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
