# Peer review of "Analytical Determination of Static Deflection Shape of an Asymmetric Extradosed Cable-Stayed Bridge Using Ritz Method"

_materials, 2022, doi:10.3390/ma15124255_

Round 1

Reviewer 1 Report

This is an interesting work, and the model and methods they developed are very solid. Hence, I suggest publishing this paper. However, some major issues should be addressed before publishing.

  1. The authors only compare their calculated results with experimental data – MEF and Ritz method. If it is possible, the authors could give some results compared with other previous literature in order to show the generality of their methods. If not, some discussion could also be added.
  2. Equations: check if you have explained all the terms that you used.
  3. References 34, 33, 30, 29, 28, 27, 26, 25, et are missing. Checkt hat all references are included in according with template paper.
  4. The results appear to be marginal to the field of scientific research considered.
  5. The section Conclusions will be point out the original results of the paper and can be extended to highlight the contributions. Please provide a clear justification for your work in this section, and indicate uses and extensions if appropriate.
  6. I think the author need to emphasize more clearly the contribution of the manuscript from a scientific point of view.

Reviewer 2 Report

 The mechanical behavior of the asymmetric extradosed cable-stayed (AECS) bridge is investigated and parametric study of AECS 19 bridge is carried out simultaneously.

Research article is well written and informative and well organized. This can be accepted in present form.

Author Response

Thanks for your approval for our paper.

Reviewer 3 Report

=== Editorial Comments ===
1) In Figure 2, please also include labels for L1, L2, L3, and L3', as they are called in Lines 128-129. This would make the figures more readable, as the authors use these terms throughout.
2) The scales for the two plots in Figure 5 should be set the same to compare deflection magnitudes for both spans.
3) Equations (1) and (2), after Line 144, should be for x1 and x2, respectively... not just x. Similar comment for other equations where x is used, for example Equation (3).
4) Please mention that Equations (1) and (2) implicitly contain boundary conditions that displacement is zero at x1 = 0 and x2 = 0. Perhaps add two new constants a0 and b0, which are both equal to zero once these boundary conditions are applied.
5) In Figure 8, it is difficult to see both lines in the plot. Please change one of the line types or boxes so both are visible.
6) In Line 220, the error in cable for ratio k is not "basically within 5%". The maximum is 7.63%

=== Technical Comments ===
1) It is asserted that the vertical component of the cable load can be treated as a vertical distributed load per Konstantakopoulos et al. (2010). However, this assumption was not specifically checked in this paper, and it seems that may have a strong impact on the computation of the cable forces (though it may have an insignificant effect on the girder deflections). Please use the FEM results to verify that this assumption is true for the investigated bridge.
2) The analysis was considered for a unit load of 1 kN/m. I assume that this was done using a linear elastic model, presuming that the structure remains linear even under design loading. If design-level loading is applied to these bridges, are material nonlinearities or large-deflection concerns (i.e., nonlinear geometry, P-delta effects, etc.) necessary to modeling the structure? If so, then the structural analysis should perhaps be conducted under the design loads rather than a unit load.
3) Does this method still work if loading is not uniformly distributed along the length of the bridge (for example, what if only one span is loaded)? How would the method be modified to account for this, and would this decrease the accuracy of the method?
4) Does the tower shorten under axial loads? If so, how could this be included in the model?
5) Please expand the methodology to examine the potential energy within the entire structure (not just the girder, but also the flexural strain energy in the tower and the axial strain energy in the cables). I feel this might address some of the inaccuracies in the current method. This new result could be compared to the results from considering only the potential energy in the girder. This might address some of the difficulties and assumptions inherent in estimating cable forces t and kt.
6) The differences between FEM and RMF in the cable forces seems significant. Why are these so different? The "fix" for this to exchange M1 for M1' = M1/1.5 seems arbitrary and unjustified - why is this necessary and physically justified (not just to make the model work out... but provide a scientific justification).
7) Follow up to comment 6: do all the results in Table 2 and the rest of the following analyses use M1' = M1/1.5?
8) Discussion of the parametric studies (section 4), does not provide enough information on how these studies were performed. Were these all starting with the same "basic model" from Section 3, and then varying some quantity? Some related notes:
a) Section 4.1 looking at varying L1 and L3, what remains constant? For example, if L1 is increased, does this increase the entire span of the bridge, or does the span length L stay the same, meaning there is a corresponding decrease in L2 while keeping L3 the same? Basically, it is not clear from the paper how the variations in a single parameter are conducted, and what other (if anything else) dependent parameters also change. Another example: when varying L3/L, does L3' stay the same? This seems to be adding extra asymmetry to the bridge.
b) It's not clear how general these findings are, or if they are specific to the base geometry chosen for the case study. For example, lines 267-268, does the span ratio of 0.7 always reverse this deflection, or does that depend somehow on the values (or ratios) of L1, L2, L3, and L3'? For different span length or perhaps cable angles, is this still true?
9) Conclusion 2 in Line 298 states that the analysis of the unsupported length of the girder "shows" that the stay cables act as a series of elastic supports on the girder. This was not actually checked anywhere in the paper (see comment 1). Furthermore, an elastic support is not the same as a uniformly distributed load t or kt as done in this paper. An elastic support implies that the magnitude of the reaction is proportional to the deflection at this point - this would result in a distributed load only if all the cables were strained equally, which was not verified.

Reviewer 4 Report

The reviewer thanks the authors for the work done. The technical contents of the paper are in general interesting. The “Lingjiang Bridge” has been used as a basic model here. Nevertheless, the publication in the “Materials, MDPI” is not recommended unless the following suggestions are taken into account within the article:

1) The title of the paper should be changed with “Analytical Determination of Static Deflection Shape of an Asymmetric Extradosed Cable-Stayed Bridge using Ritz Method”.

2) The numerical analyses illustrated by the authors refer to the uncracked concrete condition of “Lingjiang Bridge”. Please, specify within the text.

3) The finite element analyses must be better explained with, particularly, details of the model used. It is not clear how the model of the bridge is composed (beam elements, plate and shell etc., with the corresponding amounts and mesh sizes) and how the external loading were applied on it. Which are the geometric characteristics of cables and concrete bridge-girders ? Moreover, have geometric nonlinear analyses been performed ? Please, revise the section and provide more information about the model.

4) Which commercial software has been used for finite element analyses ? Please, specify and report the corresponding reference of the software.

5) The obtained displacements using Ritz method were not compared with conventional formulas. A table could report the comparison of the displacements achieved using conventional formulas, FE analyses and Ritz method.

6) Figure 1: “Configuration of the basic model”. Please, indicate the units.

7) The term “mid-span” should be changed with the term “midspan” within the text.

8) There are too many parameters described as formulas within the text. E.g., Eqs. (7) and (8) can simply be reported within the text.

9) The further work, related to this research, should be mentioned at the end of the article. The potential of the current work is for bridge applications. Will additional numerical analyses be performed on different extradosed cable-stayed bridges using Ritz Method ? Please, specify.

10) I suggest to the authors to edit all the text of the article with the help of a native English speaker. Grammar, punctuation, spelling, verb usage, sentence structure, conciseness, readability and writing style must also be improved.

Round 2

Reviewer 1 Report

I accept for publication  in present form 

Reviewer 4 Report

The authors carried out the required revisions.